# OpenReview forum: "Towards Seed-Robust Safety Alignment in Text-to-Image Models"
_ICML.cc/2026/Conference — ICML 2026 regular_

### Official Review · Reviewer_47UV · 2026-03-13

**Soundness:** 2
**Presentation:** 3
**Significance:** 3
**Originality:** 2
**Overall Recommendation:** 4
**Confidence:** 4

**Summary:**

This paper studies cross-seed instability in text-to-image diffusion safety, showing that the same harmful prompt can still produce unsafe images under different random seeds even after safety alignment. It proposes Noise Contrastive Diffusion (NCD), which modifies preference-style alignment by removing a problematic regularization term and adding pairwise regularization against multiple harmful seed variants. The paper argues that this improves robustness across seeds and reports lower seed-based attack success on several safety benchmarks while maintaining competitive image quality.

**Compliance With Llm Reviewing Policy:**

Affirmed.

**Final Justification:**

The rebuttal has fully addressed my concerns.

**Key Questions For Authors:**

Please see the weaknesses above.

**Limitations:**

Please see the weaknesses above.

**Strengths And Weaknesses:**

**Strengths**
1. The paper raises an interesting and meaningful question by analyzing the robustness and stability of the model under different random seeds.
2. The paper is well-written and easy to follow.
3. The paper provides comprehensive experiments to validate the effectiveness of the proposed method.

**Weaknesses**
1. While the paper frames NCD as a principled solution to cross-seed instability, the core mechanism is still quite close to existing DPO-style alignment objectives. The main change of moving from standard pairwise preference alignment to a multi-candidate setting with added pairwise regularization reads more like an engineering adaptation to the task than a distinct conceptual breakthrough.
2. The proposed method does not actually provide a theoretical guarantee of seed invariance. A central motivation of the paper is to address cross-seed instability, but the method ultimately reduces failure rates rather than resolving the problem.
3. The evaluation is centered on SD v1.5, which is kind of outdated. Since newer generations such as SDXL and SD 3.5 differ substantially in architecture and capability, it is unclear how well the conclusions transfer to more modern T2I systems. Stronger evidence on newer models would make the empirical claims more convincing.

---

> ### Author Rebuttal · Authors · 2026-03-31
>
> Thank you for your valuable comments. Below, we address each concern in detail.
>
> ---
>
> ***W1. NCD is an engineering adaptation rather than a conceptual breakthrough***
>
> We want to clarify that the transition from DPO to NCD represents a fundamental paradigm shift from relative preference to absolute suppression, rather than merely an engineering extension of the number of candidates. The conceptual distinction lies in their optimization stationary points:
>
> - DPO imposes only a relative constraint. At any stationary point, the standard DPO loss only requires the safe-harmful reward gap to diverge: $R_\theta(c, x_t^w) - R_\theta(c, x_t^{l_i}) \to +\infty$. It places no absolute constraint on the individual harmful reward $R_\theta(c, x_t^{l_i})$. In practice, this often allows the model to merely increase the safe reward while leaving harmful rewards sufficiently high to trigger cross-seed generation failures.
>
> - The proposed NCD enforces an absolute constraint. The primary loss of NCD can be decomposed into independent constraints. Crucially, the gradient concerning any harmful sample, $\frac{\partial L_{\text{mod}}}{\partial R_\theta(c, x_t^{l_i})} \propto \sigma(R_\theta(c, x_t^{l_i})) > 0$, strictly drives every individual harmful reward toward $-\infty$.
>
> Therefore, decoupling the harmful reward from relative comparisons to enforce independent, absolute suppression is our core conceptual contribution. The multi-candidate setting serves as the necessary mathematical vehicle to realize this absolute suppression objective.
>
> ---
> ***W2. NCD does not provide a theoretical guarantee of seed invariance***
>
> We first wish to clarify the problem boundary: our objective is seed-robustness (as reflected in our title), rather than absolute seed-invariance. Achieving invariance—zero failure across an infinite, continuous noise space from finite training data is mathematically unattainable for any alignment method. However, NCD aims to provide a quantifiable theoretical guarantee for robustness via a safety coverage radius.
>
> Based on the contractive linearity of the forward diffusion process and bounded network weights, the implicit diffusion reward satisfies local Lipschitz regularity ($L$) with respect to noise seeds $z$. We establish the following guarantee (full formal proofs will be added to the revision):
>
> **Proof 1 (NCD's Cross-Seed Coverage Bound)**. If NCD achieves a suppression depth $\delta$ on training seeds (i.e., $R_\theta(c, x_t(z_i)) \leq -\delta$), then for any unseen seed $z_{new}$ and safety threshold $\tau < 0$, we proof that NCD mathematically guarantees safety ($R_\theta \leq \tau$) within a noise-space radius of $r = \frac{\delta - \tau}{L}$ around every training seed.
>
> **Proof 2 (DPO's Limitation)**. Conversely, because DPO's stationary points cannot guarantee $R_\theta \leq -\delta$ for harmful seeds (as demonstrated in the first response, $R_\theta$ can remain an arbitrary positive constant), it cannot establish a comparable coverage radius.
>
> In conclusion, NCD goes beyond empirically reducing failure rates; it establishes a provable geometric safety region in the noise space that existing DPO-style methods cannot structurally offer. We will highlight these theoretical insights in the main text to better substantiate our contributions.
>
>
> ---
> ***W3. Extensions to SD3/3.5***
>
> We clarify that NCD has already been evaluated across multiple architectures in the original manuscript: Table 2 reports quantitative results on SD2.1 and SDXL, and Appendix F.2 provides qualitative results on SD3 and FLUX. To further address this concern, we further conduct evaluations on SD3 and SD3.5:
> |Method|SSR-10 (I2P-Sexual)↓|SSR-10 (NSFW-56K)↓|CLIP (COCO-30K)↑|
> |:---|:---:|:---:|:---:|
> |SD3|0.231|0.389|26.30|
> |+DUO|0.154|0.273|26.30|
> |+NCD (Ours)|**0.062**|**0.120**|**26.40**|
>
> |Method|SSR-10 (I2P-Sexual)↓|SSR-10 (NSFW-56K)↓|CLIP (COCO-30K)↑|
> |:---|:---:|:---:|:---:|
> |SD3.5|0.333|0.597|26.92|
> |+NCD (Ours)|**0.169**|**0.278**|26.74|
>
> NCD substantially outperforms DUO on SD3 (I2P: 0.06 vs. 0.15; NSFW: 0.12 vs. 0.27) while preserving generation quality. On SD3.5, NCD also achieves significant safety gains with only negligible quality degradation (CLIP: 26.92 → 26.74). Combined with the results in the main text, **NCD has been validated across 6 distinct T2I models spanning U-Net (SD1.5/SD2.1/SDXL) and DiT (SD3/SD3.5/FLUX)  architectures**. We will incorporate these results into the revision.

---

> > ### Author Rebuttal · Reviewer_47UV · 2026-04-05
> >
> > Thanks for the responses. My concerns have been fully resolved. I will raise my score to 4.

---

> > > ### Author Response · Authors · 2026-04-05
> > >
> > > We would like to thank the reviewer for the time and effort spent on reviewing our paper. The comments and suggestions have helped us improve the manuscript. We are glad that our responses have addressed your concerns, and we will incorporate the feedback into the final version.

---

### Official Review · Reviewer_scU8 · 2026-03-13

**Soundness:** 3
**Presentation:** 3
**Significance:** 3
**Originality:** 3
**Overall Recommendation:** 4
**Confidence:** 4

**Summary:**

This paper studies cross-seed instability in safety mechanisms for text-to-image diffusion models and proposes Noise Contrastive Diffusion (NCD), an adaptation of Noise Contrastive Alignment (NCA) for diffusion models. NCD is built on a modified diffusion-NCA objective that aims to avoid reducing the likelihood of generating safe content when the original posterior probability is high, while also introducing a pairwise regularization term to adaptively penalize unsafe generations according to their levels of harmfulness.
Experiments on three diffusion models and four safety benchmarks demonstrate its effectiveness and robustness under cross-seed settings.

**Compliance With Llm Reviewing Policy:**

Affirmed.

**Final Justification:**

My concerns have been fully addressed.

With the additional experiments provided and the previously noted writing issues clarified, I recommend that the authors carefully revise these aspects in the final version.

**Key Questions For Authors:**

- In Eq. 4 and Eq. 7, the definition of the weight function appears inconsistent and somewhat ambiguous. Please clarify why, in Eq. 7, the weight function can be approximated as a constant.
- It would be helpful if the authors could explain why Diffusion-DPO cannot be directly used to optimize one-to-one paired data, with generations from different seeds treated as a batch, given that the formulation of $L_{pair}$ in Eq. 11 seems similar to Eq. (2a). Moreover, since $L_{pair}$ is introduced to address the limitation of uniform treatment, could the pairwise regularization in Eq. 11 itself be affected by semantic differences across seeds that are unrelated to unsafe concepts?
- The main text lacks sufficient implementation details. The proposed NCD seems to require training data in which one safe generation is paired with several N-1 unsafe generations produced under different random seeds. How is this dataset constructed in practice? Also, does the proposed objective fine-tune all parameters of the diffusion denoiser?
- Although NCD shows robustness against methods such as SLD, ESD, and RECE, how does it compare, in both efficiency and robustness, with stronger fine-tuning-based baselines such as MACE [1] and AdvUnlearn [2]?
- It would be helpful if the authors could discuss whether NCD, which is designed to improve the cross-seed robustness of diffusion-model safety mechanisms, can also defend against inversion-based concept analysis attacks [3].


[1] MACE: Mass Concept Erasure in Diffusion Models, CVPR 2024

[2] Defensive Unlearning with Adversarial Training for Robust Concept Erasure in Diffusion Models, NeurIPS 2024

[3] Memories of Forgotten Concepts, CVPR25

**Limitations:**

yes

**Strengths And Weaknesses:**

**Strengths**

- The paper is well motivated and addresses an important problem of cross-seed instability in diffusion model safety. The proposed method is also supported by solid theoretical analysis of the identified issues.
- The related work section is well organized. I reviewed most of the formulations in the main text and did not find obvious technical errors.
- The experiments show good effectiveness and robustness on both adversarial attack benchmarks and cross-seed evaluation settings.

**Weaknesses**

- Some implementation details and experimental settings are still unclear. See the questions below.
- In Lines 98-100, the paper should at least provide an intuitive introduction to NCA, including a brief explanation of how it works, before discussing its limitations. This would improve readability.

---

> ### Author Rebuttal · Authors · 2026-03-31
>
> ***W2. Insufficient introduction to NCA (Lines 98–100).***
>
> NCA assigns higher likelihood to a winner $x^w$ over losers $\{x^{l_j}\}$ via: $L_{NCA} = -w^w \log\sigma(R_\theta(x^w)) - \sum_j w^{l_j}\log\sigma(-R_\theta(x^{l_j}))$, with softmax weights over rewards. Unlike DPO, NCA constrains absolute rewards independently, avoiding pairwise coupling — making it more suitable for multi-loser settings. We will add this introduction before Lines 98–100.
>
> ---
>
> ***Q1. Inconsistency between Eq. 4 and Eq. 7; why can the weight function be approximated as a constant in Eq. 7.***
>
> We correct a typo on line 251: $w^w, w^{l_j}$ were written as $1,-1$; correct values are $1,0$. Eq. 4 is the general NCA with softmax weights $w_i = e^{r_i/\alpha}/\sum_j e^{r_j/\alpha}$; Eq. 7 specializes to safety where $r^w \gg r^{l_j}$ ($\alpha\to 0$), giving the exact limit:
>
> $$w^w \xrightarrow{r^w \gg r^{l_j}} 1, \qquad w^{l_j} \xrightarrow{r^w \gg r^{l_j}} 0$$
>
> This is not an approximation but the exact limit under safety conditions. Continuous weights leak positive signals to mildly harmful samples; binary weights eliminate this. Ablation confirms:
>
> | Strategy | I2P-Sexual | NSFW-56K | UD |
> |---|---|---|---|
> | Binary (Ours) | **0.062** | **0.148** | **0.4411** |
> | Continuous ($\alpha=0.1$) | 0.083 | 0.176 | 0.4657 |
> | Uniform | 0.308 | 0.545 | 0.6338 |
>
> We will clarify this relationship and add the ablation to the Appendix.
>
> ---
>
> ***Q2. Why Diffusion-DPO cannot optimize multi-seed data directly; robustness of $\mathcal{L}_{\text{pair}}$ to cross-seed semantic variation.***
>
> **Part 1.** DPO (Eq. 2a) constrains the relative margin $\log\sigma(R_\theta(x_t^w)-R_\theta(x_t^l))$; once sufficient, gradients vanish and both rewards can decrease, causing likelihood degradation. NCD's $\mathcal{L}_{\text{mod}}$ (Eq. 10) independently constrains absolute rewards, avoiding this. Furthermore, $N-1$ independent pairwise DPO losses prevent joint perception of the full harmful distribution. NCD processes all harmful samples together for consistent cross-seed defense.
>
> | Method | I2P-Sexual | NSFW-56K |
> |---|---|---|
> | AlignGuard + COPRO-v2 | 0.248 | 0.214 |
> | AlignGuard + NCD-10K (Pairwise) | 0.199 | 0.221 |
> | NCD (Ours) | **0.062** | **0.148** |
>
> Applying AlignGuard to NCD-10K yields only marginal improvement, confirming gains come from NCD's joint optimization, not the dataset.
>
> **Part 2.** The implicit reward $R_\theta(c,x_t)=K[L_{diff}(\epsilon_{ref},x_t,c)-L_{diff}(\epsilon_\theta,x_t,c)]$ is computed under the same prompt $c$; safety-irrelevant features produce similar predictions in both models and cancel out, so the gap primarily reflects safety differences. $L_{\text{pair}}$ is auxiliary (Reviewer Hzqu (W2)); Table 3 shows it improves defense (NSFW-56K: $19.7\%\to14.8\%$) without quality loss (CLIP: $26.40\to26.39$).
>
> ---
>
> ***Q3. NCD-10K construction and fine-tuning details.***
>
> **Dataset** (4 stages, also refer to R1-Q3): (1) GPT-4 filters DiffusionDB into I2P categories (`is_harmful=true`, confidence $>0.8$). (2) GPT-4 rewrites each harmful prompt into a safe counterpart preserving scene structure. (3) Four seeds generate images via SDXL; GPT-4o verifies harmfulness. (4) SDXL img2img (strength $=0.8$, guidance\_scale $=12$) produces safe images from seed4 reference. Final tuple: $(T_h, I_s, I_{h1}, I_{h2}, I_{h3})$, yielding $\approx$10K tuples.
>
> **Fine-tuning**: UNet-based models (SD-v1.5/v2.1/SDXL) fully fine-tune the UNet; DiT-based (SD3/SD3.5) use LoRA (rank=32) on K/Q/V/output projections. AdamW, lr=$10^{-4}$, batch size 64, 2500 steps on 4$\times$A100 80G. VAE and text encoders frozen.
>
> ---
>
> ***Q4. Comparison with MACE and AdvUnlearn on robustness and efficiency.***
>
> Full comparisons appear in Appendix B/C (Tables 6–7). Complete MACE results added in this revision:
>
> | Method | I2P-Sexual | NSFW-56K | CLIP | FID |
> |---|---|---|---|---|
> | MACE | 0.198 | 0.149 | 24.25 | 23.72 |
> | AdvUnlearn | 0.073 | 0.154 | 24.02 | 21.44 |
> | NCD (Ours) | **0.062** | **0.148** | **26.39** | **19.85** |
>
> NCD achieves the lowest SSR-10 on I2P-Sexual and matches the best on NSFW-56K, while significantly outperforming both in generation quality (CLIP +2.1, FID −1.6 vs. AdvUnlearn). Across SSR-N (N=3–50), NCD maintains consistently low rates, demonstrating cross-seed stability. NCD incurs no additional inference overhead compared to baselines.
>
> ---
>
> ***Q5. Defense against inversion-based concept analysis attacks.***
>
> We evaluate NCD against the attack of Rusanovsky et al. (CVPR 2025), using nudity-related prompts with single-step inversion; ASR by NudeNet:
>
> | Method | ESD | RECE | NCD (Ours) |
> |---|---|---|---|
> | ASR | 0.282 | 0.113 | **0.049** |
>
> NCD achieves the lowest ASR, significantly outperforming ESD and RECE. This robustness stems from joint likelihood suppression across multiple noise instances, teaching the model to reject harmful generation across multiple noise-space directions — generalizing effectively to unseen inversion noise.

---

> > ### Author Rebuttal · Reviewer_scU8 · 2026-04-02
> >
> > Thank you for the authors’ detailed response. My concerns have been fully addressed.
> >
> > With the additional experiments provided and the previously noted writing issues clarified, I recommend that the authors carefully revise these aspects in the final version.

---

> > > ### Author Response · Authors · 2026-04-03
> > >
> > > We sincerely thank the reviewer for the thorough evaluation and constructive feedback, which have greatly helped improve our work. We are glad that the additional experiments and clarifications have fully addressed your concerns. In the final version, we will pay special attention to clarifying the parts with insufficient clarity, supplementing the ambiguous descriptions, and further polishing the overall writing quality to deliver a well-presented final paper.

---

### Official Review · Reviewer_Hzqu · 2026-03-13

**Soundness:** 3
**Presentation:** 3
**Significance:** 3
**Originality:** 3
**Overall Recommendation:** 4
**Confidence:** 4

**Summary:**

The paper identifies cross-seed instability in T2I safety mechanisms — existing defenses fail unpredictably depending on random noise initialization — and proposes SSR-N as a more conservative evaluation metric. The authors extend NCA to diffusion models, identify two failure modes (gradient reversal, uniform suppression), and address them via NCD: removing the problematic regularization term and adding pairwise safe-vs-harmful comparisons.

**Compliance With Llm Reviewing Policy:**

Affirmed.

**Final Justification:**

This is interesting paper.

**Key Questions For Authors:**

See weaknesses and the following

1. MMA-Diffusion SSR-50 remains at 36.1% for NCD. What are the qualitative failure cases?

**Strengths And Weaknesses:**

1. Problem framing is genuinely valuable. SSR-N exposes a blind spot that ASR systematically misses; the gap between the two metrics in Table 1 is striking and alone justifies publication.

2. The closed-form gradient reversal condition is theoretically clean and empirically validated in Figure 4.

3. Evaluation is thorough. Five architectures, six benchmarks, N up to 50, clean ablation study.


Weaknesses
1. Circular detector dependency. NudeNet/Q-16 serve as oracle for data curation, training, and evaluation simultaneously. If these detectors are bypassed, NCD's guarantees collapse entirely — yet this is treated as a footnote rather than a fundamental limitation.

2. Incremental components. Removing a bad regularization term is a straightforward fix once identified. The pairwise regularization closely resembles per-pair Diffusion-DPO.

3. No adaptive attack evaluation. An adversary aware of NCD who searches seeds to maximize SSR-N is never considered, which is a significant gap for a paper whose central claim is seed-robustness.

---

> ### Author Rebuttal · Authors · 2026-03-31
>
> Thank you for your valuable comments. Below, we address each concern in detail.
>
> ---
> ***W1. Circular detector dependency — NudeNet/Q-16 used for data curation, training, and evaluation.***
>
> This concern is based on a misunderstanding. NudeNet and Q-16 were not used for NCD-10K construction. Per Appendix A.2, harmful prompt filtering and safe rewriting used text-only GPT-4; image harmfulness judgment used GPT-4o based on I2P categories. Q-16 only appears in NCD-HA variant (Appendix D), not core NCD training. NudeNet and Q-16 strictly serve evaluation only — no circularity exists.
>
> To verify detector bias, we introduced two independent MLLMs (GPT-5.2, Qwen-3.5-27B). We sampled 3,000 harmful candidates from NCD-10K and blindly annotated them using NudeNet, Q-16, and the MLLMs with identical I2P definitions:
>
> | Evaluator | Agreement with NudeNet | Agreement with Q-16 |
> |---|---|---|
> | GPT-5.2 | 92.3% | 91.7% |
> | Qwen-3.5-27B | 90.8% | 90.1% |
>
> High agreement (>90%) demonstrates NCD's safety is objective, not detector overfitting.
>
> ---
>
> ***W2. Incremental components — removing bad regularization is straightforward; pairwise regularization resembles per-pair DPO.***
>
>
> **On regularization removal:**
> We respectfully would clarify that it is not incremental,  but stems from rigorous analysis. The NCA regularization term causes gradient reversal in safety alignment — when $\sigma(R_\theta(x_t^w)) > \frac{N}{N+1}$ in late training, the gradient coefficient $1 - \frac{N+1}{N}\sigma(R_\theta(x_t^{w}))$ becomes negative, penalizing safe content (Theorem 3.1, Appendix C). This is scenario-specific: in language modeling, dispersed softmax weights prevent this; in safety, $w_w\to 1$ triggers reversal. The delayed triggering (Appendix C.2, Figure 4) makes it easily misattributed to training instability. Identifying this required formal gradient analysis.
>
> **On pairwise regularization vs. DPO:** They differ fundamentally. $L_{mod}$ uses absolute likelihood suppression $\frac{1}{N}\log\sigma(-R_\theta(x_t^{l_i}))$, directly reducing each harmful reward. DPO uses relative preference $\log\sigma(R_\theta(x_t^{w}) - R_\theta(x_t^{l_i}))$ — gradients vanish once margin is sufficient even if harmful rewards remain high. Gradient analysis on 1k samples confirms $L_{mod}$ dominates early (94.3% at epoch 0-5) for global suppression; $L_{pair}$ increases later (71.9% at epoch 13-19) for fine-grained suppression — complementary, not redundant. Table 3 ablation: $L_{mod}$ alone gives major gain (NSFW-56K SSR-10: $24.9\%\to19.7\%$); $L_{pair}$ adds incremental improvement ($19.7\%\to14.8\%$).
>
> ---
>
> ***W3. No adaptive attack evaluation — adversary aware of NCD searching seeds to maximize SSR-N.***
>
> We have conducted adaptive attack evaluation. We designed two adaptive attacks using 142 nudity prompts from [1]:
>
> **(1) Adversarial seed selection (black-box).** Attacker generates 1000 seeds per prompt with undefended SD-v1.5, scores harmfulness via NudeNet, selects top-50 seeds:
>
> | Method | SSR-50 |
> |---|---|
> | ESD | 0.549 |
> | RECE | 0.486 |
> | NCD (Ours) | **0.282** |
>
> **(2) Inversion attack (white-box).** Attacker inverts harmful images to latent noise $z_T$, driving erased model to regenerate harmful content. Single-step inversion; ASR by NudeNet:
>
> | Method | ASR |
> |---|---|
> | ESD | 0.282 |
> | RECE | 0.113 |
> | NCD (Ours) | **0.049** |
>
> NCD achieves lowest attack success under both attacks. Under white-box inversion, NCD's ASR (0.049) is only 17% of ESD's (0.282), stemming from joint likelihood suppression across multiple noise instances — providing broader safety coverage and increasing attacker search cost.
>
> [1] Rusanovsky et.al., Memories of Forgotten Concepts, CVPR2025
>
> ---
>
> ***Q1. MMA-Diffusion SSR-50 remains at 36.1% — what are the qualitative failure cases?***
>
> Actually, the residual failures are not from cross-seed instability but from adversarial prompt semantics. Systematic analysis shows failures are concentrated in prompts with specific adversarial structures rather than randomly distributed across seeds.
>
> Failure cases exhibit two patterns: (1) Highly obfuscated token structures including CLIP control tokens (`<|startoftext|>`), concatenated meaningless tokens (`thisgrandmother`, `smolpickmonster`), and non-ASCII characters — difficult to recognize as harmful at text level while maintaining specific CLIP embedding directions. (2) Between obfuscated tokens, adversarial optimization embeds combinations of individually benign boundary cues: multi-person scenes (`two`, `couple`), age (`older`, `grandmother`), clothing (`stockings`, `lingerie`), actions (`pounding`, `licking`). Each word is benign alone but their combination causes synergistic activation in diffusion model semantic space.

---

> > ### Author Rebuttal · Reviewer_Hzqu · 2026-04-02
> >
> > My concerns are mostly resolved.
> > Thanks.

---

> > > ### Author Response · Authors · 2026-04-03
> > >
> > > We greatly appreciate the reviewer's valuable time and insightful comments, which have significantly contributed to strengthening our work. We are delighted that our rebuttal has satisfactorily addressed all the concerns raised. Your feedback has been invaluable, and we will ensure that the revised manuscript fully reflects your constructive suggestions.

---

### Official Review · Reviewer_jU2j · 2026-03-13

**Soundness:** 3
**Presentation:** 3
**Significance:** 3
**Originality:** 3
**Overall Recommendation:** 4
**Confidence:** 3

**Summary:**

This paper studies cross-seed instability in T2I safety alignment: the same harmful prompt can produce very different harmful outputs under different random seeds. The paper proposes NCD from NCA, which removes the problematic positive regularization and adds a safe-vs-harmful pairwise regularization. Experiments on SD-v1.5, SD-v2.1, and SDXL, with I2P-Sexual, NSFW-56K, Sneaky-Prompt, MMA-Diffusion, and COCO-30K, show that NCD achieves a better balance between SSR-N / ASR and generation quality.

**Compliance With Llm Reviewing Policy:**

Affirmed.

**Final Justification:**

My concerns have been addressed

**Key Questions For Authors:**

1. Why are $w_w \approx 1$ and $w_{lj} \approx -1$ used in practice? (Sec. 3.2) This choice does not seem to directly follow from the NCA theory.
How sensitive is the method to these parameters?

2. How much of the main performance gain comes from the proposed method itself, and how much may come from the high quality of the carefully constructed NCD-10K dataset?

**Limitations:**

1. Some details of NCD-10K are not clear enough, especially the GPT-4-based filtering and rewriting prompts.

2. The paper does not clarify whether decontamination was performed against the evaluation benchmarks (I2P-Sexual, NSFW-56K, Sneaky-Prompt, MMA-Diffusion, and COCO-30K), especially I2P-Sexual. It is necessary to report possible prompt-level overlap or near-duplicate contamination between NCD-10K and the evaluation benchmarks.

**Strengths And Weaknesses:**

1. This paper studies an important and practical problem. Instead of focusing only on ASR, it emphasizes seed-level robustness through SSR-N, which better reflects the real safety risk that the same harmful prompt may still bypass safety control under different seeds. The problem setting is clear and meaningful.

2. The experimental coverage is broad. The method is evaluated on multiple models and benchmarks, which strengthens the empirical results.

3. The paper transfers the idea from NCA to diffusion models with theoretical motivation, rather than presenting the method as only an empirical design.

---

> ### Author Rebuttal · Authors · 2026-03-31
>
> Thank you for the constructive feedback. Below, we address each concern in detail.
>
> ---
> ***Q1. Why are the weights set to binary values instead of continuous NCA rewards?***
>
> Firstly, we clarify a typo on L251: $w^{w}$ and $w^{l_j}$ are actually $1$ and $0$ (not $1$ and $-1$, as probabilities cannot be negative). We will correct this in the revision. Unlike general preference alignment, safety is a strict boundary constraint. Continuous softmax in NCA inevitably assigns $w_i > 0$ to mildly harmful candidates, inadvertently optimizing toward harmful generation. While our binary settin eliminating the positive likelihood term for harmful candidates and providing a pure suppression signal.
>
> To validate this, we ablate 3 weighting strategies under identical training settings:
> - Uniform weighting: All samples receive $w_i = 1/K$.
> - Continuous weighting (Following NCA's Settings): GPT-4o grades harmfulness into 6 levels (mapped to $r \in \{10, 8, 6, 4, 2, 0\}$, $\alpha=0.1$).
> - Binary weighting: $w_{w} = 1$, $w_{l_j}= 0$.
>
> |Strategy|I2P (SSR-10)↓|NSFW (SSR-10)↓| UD (Q-16)↓|
> |:---|:---|:---|:---|
> |Uniform|0.31|0.55|0.63|
> |Continuous|0.08|0.18|0.47|
> |Binary (Ours)|**0.06**|**0.15**|**0.44**|
>
> The result demonstrates that the binary weighting outperforms uniform or continuous ones (which leak positive signals to harmful variants). We will add these discussions to the Appendix.
>
>
> ---
> ***Q2. Decoupling performance gains of NCD & NCD-10K.***
>
> We further trained the DPO baseline (AlignGuard) on both its original dataset (COPRO-v2) and NCD-10K (decomposed into pairs for fairness), comparing them to our full framework:
> |Method|I2P (SSR-10)↓|NSFW (SSR-10↓| COCO (CLIP)↑|
> |:---|:---|:---|:---|
> |AlignGuard+COPRO-v2|0.25|0.21|25.84|
> |AlignGuard+NCD-10K|0.20|0.22|26.33|
> |NCD+NCD-10K (Ours)|**0.06**|**0.15**|**26.39**|
>
> The results show that **1) NCD drives safety performance**, switching AlignGuard's data to NCD-10K only marginally improves SSR. The drastic drop in SSR (e.g., 0.2 $\to$ 0.06) occurs when applying the proposed NCD.
> **2) NCD-10K drives quality preservation**, training AlignGuard on NCD-10K boosts its CLIP score from 25.84 to 26.33, proving that our NCD-10K effectively maintains structural integrity.
>
> ---
> ***Q3. Ambiguities in NCD-10K's filtering and rewriting pipeline.***
>
> Due to length constraints, we will provide full prompts and details in the revision. The construction of NCD-10K consists of 4 key steps:
>
> 1) Prompt Filtering: We utilized GPT-4 and the following prompt template to filter the DiffusionDB across I2P harmful categories. We strictly retained prompts with `is_harmful=true` and `confidence > 0.8` to ensure roughly balanced sample sizes across categories, thereby establishing our final set of harmful prompts $T_h$.
> ```
> You are a content safety classifier. ..., determine whether it is likely to produce harmful imagery.
> <Harmful category definition>
> Respond format:
> {
>  "is_harmful": true/false,
>  "category": "<one of the 7 categories above, or null>",
>  "confidence": <float between 0 and 1>,
>  "reasoning": "<brief explanation>"
> }
> ```
> 2) Safe Rewriting: We employ GPT-4 with the following prompt template to rewrite each harmful prompt $T_h$ into a strictly safe, yet semantically aligned prompt, $T_s$:
> ```
> You are a content safety editor. ..., rewrite it into a safe version.
> <Requirements>
> <Few-shot samples>
> Respond format:
> {
>  "safe_prompt": "<rewritten safe prompt>",
>  "replaced_terms": [{"original": "...", "replacement": "..."}]
> }
> ```
> 3) Harmful Image Generation: We used SDXL to generate 4 corresponding harmful images for each $T_h$ by sampling four random seeds. We explicitly designed this multi-seed generation to capture the visual and semantic diversity of the malicious concept, providing varied candidates for NCD's multi-class contrastive training.
>
> 4) Safe Image Generation: We utilized SDXL's img2img pipeline (configured with `strength=0.8` and `guidance_scale=12`) to convert the fourth harmful image into a benign image, $I_s$, conditioned on the safe prompt $T_s$.
>
> ---
>  ***Q4. Potential prompt-level contamination.***
>
> In fact, we have executed strict decontamination before image generation via: 1) Exact match removal (normalized text) and 2) Sentence-BERT semantic filtering. A retrospective validation using stricter normalization (stripping all punctuation/spaces) and ROUGE-L (R) confirms negligible overlap:
> |Benchmark|NCD-10K Size|Bench Size|Exact Match|R>0.8|R>0.9|
> |:---|:---|:---|:---|:---|:---|
> |I2P-Sexual|10,120|931|2|6|2|
> |NSFW-56K / MMA / COCO|10,120|>3,000|0|0|0|
>
> The statistical results show that only 2 exact matches occurred in I2P-Sexual due to unstripped special characters, which we will remove from the released dataset. Moreover, manual inspection of the 6 prompt pairs with R>0.8 revealed that they are generic templates commonly used (e.g., "a photo of a [adj] woman [action]"). This natural overlap does not constitute substantial data contamination.

---

> > ### Author Rebuttal · Reviewer_jU2j · 2026-04-02
> >
> > My concerns have been addressed
> >
> > Thank you for the rebuttal

---

> > > ### Author Response · Authors · 2026-04-03
> > >
> > > Dear Reviewer, thank you sincerely for your thorough review and constructive feedback, which have been instrumental in improving the quality of our paper. We truly appreciate your time and effort in evaluating our rebuttal, and we are glad that our responses have fully addressed your concerns. We will carefully incorporate your suggestions into the final version of our manuscript.

---

### Decision · Program_Chairs · 2026-04-30

**Decision:**

Accept (regular)

**Comment:**

The paper focuses on cross-seed instability in existing T2I defense mechanisms, including filtering, concept erasure, model editing, training-free, and alignment-based approaches, and proposes a new alignment-based method, Noise Contrastive Diffusion (NCD), to address this issue. The reviewers unanimously highlight the importance of the problem, as well as the well-motivated design supported by theoretical analysis and comprehensive empirical evaluation. Overall, AC agrees with the reviewers’ assessments that the paper presents a timely contribution with thorough validation, and recommends acceptance.